# Impact of Natural Microorganisms on the Removal of COD and the Cells Activity of the *Chlorella* sp. in Wastewater

Qingnan Sun [1], Xiaoping Zhang [1,2,3,4,5,*] and Xin Zhang [1]

[1]  Guangzhou Higsher Education Mega Centre, School of Environment & Energy, South China University of Technology, Guangzhou 510006, China; 15344079630@163.com (Q.S.); 15198155069@163.com (X.Z.)
[2]  The Key Laboratory of Pollution Control and Ecosystem Restoration in Industry Clusters of Ministry of Education, Guangzhou 510006, China
[3]  Guangdong Provincial Key Laboratory of Solid Wastes Pollution Control and Recycling, Guangzhou 510006, China
[4]  Guangdong Provincial Engineering and Technology Research Center for Environmental Risk Prevention and Emergency Disposal, Guangzhou 510006, China
[5]  Guangdong Suchun Environmental Protection Technology Co., Ltd., Dongguan 523000, China
*   Correspondence: xpzhang@scut.edu.cn; Tel.: +86-13678920429

**Abstract:** In the treatment of wastewater containing only chemical oxygen demand (COD) by *Chlorella* sp., the cell activity and proliferation ability of *Chlorella* sp. decreased with the culture time, which affected the removal of COD in wastewater. To solve these problems, the *Chlorella* sp.—natural microorganism symbiosis system was prepared. The system was used to explore how natural microorganisms affect the cell activity and the proliferation ability of *Chlorella* sp. in wastewater. In the treatment of COD by *Chlorella* sp., the removal rate of COD decreased from 45.47% to 28.88%, with a decrease in the cell activity and proliferation ability of *Chlorella* sp. In the *Chlorella* sp.–natural microorganism symbiotic system, the removal rate of COD reached 45.75%. With the introduction of natural microorganisms, the circulation of $CO_2$ and $O_2$ between *Chlorella* sp. and natural microorganisms promoted photosynthesis and respiration, which enhanced the cell activity of *Chlorella* sp. Under the condition that the dosage of natural microorganisms was between 1% and 6%, the concentration of *Chlorella* sp. was close to the logarithmic growth phase, which maintained the proliferation ability of *Chlorella* sp. At the same time, the natural microorganisms grew and proliferated in wastewater containing only COD through preying on *Chlorella* sp.

**Keywords:** *Chlorella* sp.; natural microorganisms; symbiosis system; cell activity; predation





## 1. Introduction

In recent years, the breeding industry of China has grown dramatically [1], and China has become the largest meat consumer around the world [2]. The amounts of various pollutants produced during livestock farming has increased. A large quantity of animal feces [3], animal medicine, flushing wastewater and slaughtering wastewater [4] accumulate in the livestock and poultry breeding wastewater. These pollutants put great pressure on the human living environment. Various indicators need to be paid attention to in the wastewater treatment process, especially high concentrations of chemical oxygen demand (COD) [5,6].

Different technologies have been used in the treatment of COD, including membrane treatment, flocculation process, electrochemistry, advanced oxidation method and microbial treatment [7–11]. The microbial method has been valued because of its low input, high energy recovery and no secondary pollution. Chen et al. used a lab-scale up-flow anaerobic sludge blanket (UASB) to obtain the removal rates of TN and COD, which were about 85% and 56.5% in the breeding wastewater [12]. In the treatment, the C/N was an important influencing factor. Chen et al. operated the moving bed biofilm reactor (MBBR) for breeding



wastewater, which also confirmed this view [13]. The study of airlift MBBR showed that the mechanisms of COD removal were oxidation and microbe proliferation, and the removal rate was 50.5% [14]. In this process, COD is converted into carbon dioxide, water and the component of microorganisms. In the sequential batch reactor (SBR), the food-to-microorganism ratio showed an important effect under different COD concentrations of the high-strength organic wastewater [6]. Algae have the advantages of photosynthesis to produce oxygen, no need for external aeration and can produce high-value lipids, so their application in wastewater treatment has attracted great attention [15,16]. The COD, BOD, TN and TP were removed by *Chlorella sorokiniana* AK-1 in piggery wastewater with the removal rates of 95.7%, 99%, 94.1% and 96.9%, respectively [17]. Most of the organic matter was converted to microalgal biomass [18]. This can be used for the generation of bioenergy [19]. However, the algae system still has many drawbacks when dealing with high concentrations of wastewater. The removal of N and P, COD, BOD and heavy metals was researched in the microalgae treatment for technology optimization, the performance showing that immobilization may be a new way to develop algae applications [20]. The removal of COD by *Chlorella* sp. could obtain a higher removal rate, but the long-term cultivation and proliferation of algae led to a decrease in algae activity in the culture solution and wastewater. The system of microalgae–bacteria was the new choice to improve the removal rate. However, the influences of algae activity and the new system have been less studied in wastewater treatment.

Microalgae and other microorganisms can coexist in wastewater. Microalgae absorb carbon dioxide and organic pollutants through photosynthesis, release oxygen and produce organic matter that makes up algal cells for growth and reproduction. Bacteria consume oxygen and organic matter through respiration at the same time. The cycle of carbon dioxide and oxygen can be achieved between the algal and bacteria. The algal and bacteria work together to improve the ability to treat wastewater [21]. Under light conditions, the algal–bacterial granular sludge system was prepared in SBR, the lipid content increased significantly with the growth of the algae and partial nitrification was achieved by the bacteria [22]. Algae, anaerobic microorganisms and aerobic sludge coexist in biofilm reactors to achieve biomass growth and the removal of organic matter [18]. The use of sodium alginate (SA) immobilized the algae–bacteria system, and the use of sodium hypochlorite to pretreat wastewater can improve the growth environment of microorganisms, thereby increasing the removal rate of pollutants, but excessive sodium hypochlorite will affect the activity of microorganisms [23]. Different sizes of culture vessels have different effects on the process of algae–bacteria symbiosis treatment of nitrogen, and open containers will introduce natural microorganisms from the surroundings [24]. The new microorganisms can affect the removal effect of wastewater, and the new system may be stronger than the treatment ability of the original microbial flora [24,25].

Microalgae play important roles in water environment, but large amounts of algae can produce a series of harmful secretions, excretions and microbial debris. All of these will change the pH, disrupt the balance of the microbial community and harm the water environment. In order to effectively control the disorderly growth of algae, different technologies are used in the treatment of algae cell removal. Immersed microfiltration membranes removed more than 99% of algal cells after 180 min [26]. The problems of fouling and material reuse in the physical filtration limit the application of the membrane method. Ma et al. prepared a novel covalently bonded coagulant, CAMF, and the $CAMF_{0.6}$ removed more than 96% chla at the dosage of 40 mg/L [27]. The flocculation in the physicochemical method receives an excellent removal rate with less material input but may produce secondary pollution. Advanced oxidation has played a great role in the removal of algae. Chen et al. prepared CeMOFs to oxidize algae, and the removal ability reached $1.04 \times 10^7$ cells/mg by the new nano-superstructure MOFs with high efficiency and selectivity [28]. In the algae treatment by ozone microbubble-enhanced air flotation technology, Wang et al. demonstrated a good effect by the mechanisms of flocculation adhesion and oxidation [29]. After the microalgae are oxidized, the cell concentration is greatly reduced,

and no new pollutants appear in the wastewater. However, the prices of various oxidizing materials limit the feasibility of large-scale applications of oxidation methods. Compared to other methods, the use of biological competition to control algae is worth investigating [30]. Biological methods of algae removal include direct invasion by other microorganisms, the effects of plants extracts and the destruction of toxic substances [31]. The biological methods are more eco-friendly than other methods, with no secondary pollution. Microorganisms can survive in water for a long time due to their own growth rules, so microbial treatment can ensure that the algae concentration is at a low level for a long time. The acquisition of microorganisms is relatively simple, the source is varied and the best result can be obtained with low input, so it can be applied in large-scale processing.

The aims of this paper are (1) the preparation of a *Chlorella* sp.–natural microorganism's symbiotic system through the introduction of natural microorganisms from the environment, (2) the determination of the effects of *Chlorella* sp. culture time and extraction concentration on *Chlorella* sp. cell activity and proliferation capacity, and the enhancement of the cell activity of *Chlorella* sp. by using *Chlorella* sp.–natural microorganism's symbiotic system and (3) the control of the cell concentration of *Chlorella* sp. through the predatory action of natural microorganisms.

## 2. Materials and Methods

### 2.1. Microorganisms and Culture Conditions

*Chlorella* sp. (FACHB-28) and BG-11 medium were obtained from the Freshwater Algae Culture Collection at the Institute of Hydrobiology (FACHB), National Aquatic Biological Resource Center, Wuhan, China. The *Chlorella* sp. cell concentration was more than $1.00 \times 10^7$ cell/mL. The culture medium was BG-11 medium. The *Chlorella* sp. was cultivated in a 250 mL triangular flask in a constant temperature light incubator (Huangshi Hengfeng Medical Equipment Co., Ltd., Huangshi, China). The light source was an energy-saving lamp, and the light intensity was 2000 Lux and on a 12 h:12 h light:dark cycle; the temperature was set to 25 °C, and the flask was shaken evenly twice a day.

Firstly, isolate the air to cultivate the *Chlorella* sp., recorded as the seal group. Secondly, expose the culture solution to air, recorded as groups I and II. Compare the effect of external air on *Chlorella* sp. growth and reproduction. Finally, 100 mL of *Chlorella* sp. solution was configured, recorded as groups III and IV, and the growth curve of *Chlorella* sp. was plotted. The difference between I, II, III and IV was the initial concentration of *Chlorella* sp. (I: $1.35 \times 10^6$ cell/mL, II: $2.35 \times 10^6$ cell/mL, III: $1.01 \times 10^6$ cell/mL and IV: $2.82 \times 10^6$ cell/mL).

Natural microorganisms were introduced from the environment, and after the concentration was stable around $5.60 \times 10^6$ cell/mL, the cells were stored in 1000 mg/L COD solution.

### 2.2. Cyclic Culture of Chlorella sp.

When the cell concentration of *Chlorella* sp. reached the standard concentration ($1.60 \times 10^7$ cell/mL) in BG-11, the cell solution was extracted for centrifugation concentration, and then, the cells were added to the wastewater. The dosage was recorded as the solution volume ratio. The cell concentration in BG-11 was diluted due to the new BG-11 supplementation, and *Chlorella* sp. continued to divide and proliferate. By repeating the operation, new *Chlorella* sp. cells were continuously obtained.

### 2.3. Wastewater Treatment

The wastewater was designed to simulate the concentrations of COD in breeding wastewater. The concentrations of COD in most real breeding wastewater are between 500 and 5000 [11,12,17,32]. In this study, 2000 mg/L COD was obtained using glucose (Shanghai Aladdin Biochemical Technology Co., Ltd., Shanghai, China). The microorganisms solution was centrifuged at 3000 r/min for 10 min to obtain the cells, and then, the cells were added into the COD wastewater. The dosages of microorganisms in wastewater were recorded as the volume ratio of the solution.

All treatments were performed in 250 mL Erlenmeyer flasks, with 100 mL of COD wastewater as described. The COD removal experiments in wastewater by microorganisms were carried out at room temperature (25 ± 0.5 °C). The light source was an energy-saving lamp, the light intensity was 2000 Lux and the light incubator was set to a 12 h:12 h light:dark cycle. The Erlenmeyer flasks were shaken evenly twice a day. The experiments were performed under conditions as follows (the standard concentration (100% [*v/v*]) of *Chlorella* sp. was $1.60 \times 10^7$ cell/mL, and the standard concentration (100% [*v/v*]) of the natural microorganisms was $5.60 \times 10^6$ cell/mL): (1) wastewater containing 10%, 20% and 30% (*v/v*) cells of *Chlorella* sp.; (2) wastewater containing 30% (*v/v*) cells of *Chlorella* sp. from different batches; (3) wastewater with 1000 mg/L and 2000 mg/L COD containing 1%, 2% and 3% (*v/v*) cells of natural microorganisms and (4) wastewater containing both *Chlorella* sp. and natural microorganisms. In the treatment of wastewater by *Chlorella* sp. and natural microorganisms, the dosages of the natural microorganisms were 1%, 3% and 6% (*v/v*), respectively, and the dosages of *Chlorella* sp. were 30%, 60%, 100% and 200% (*v/v*), respectively. The dosages of *Chlorella* sp. from BG-11 were recorded as 30% BG and 60% BG, while the others were from the wastewater containing COD only. The experiments lasted 7 days, and samples were collected at 1 day intervals; at the same time, the microorganisms were counted.

### 2.4. Analytical Methods

2.4.1. Microorganisms Growth

A hemocytometer (Shanghai Aladdin Biochemical Technology Co., Ltd., Shanghai, China) were used to determine the biomass changes in *Chlorella* sp. and the natural microorganisms. Count the cells on the five medium lattices of the hemocytometer and calculate the cell concentration using Equation (1):

$$C = (l_1 + l_2 + l_3 + l_4 + l_5) \times 5 \times 10000 \text{ (cell/mL)} \tag{1}$$

where $l_i$ is the number of cells in the middle lattice $i$.

The cell characteristics of *Chlorella* sp. and natural microorganism were observed using microscopes.

2.4.2. COD Concentration

All the wastewater samples were filtered through a cellulose acetate membrane filter (0.22 μm). Pipette 3 mL of the liquid to be measured into the cuvette tube, and add 1 mL of 10% (*v/v*) $H_2SO_4$ (Shanghai Aladdin Biochemical Technology Co., Ltd., Shanghai, China), 3 mL of digestion solution (9.80 g/L $K_2Cr_2O_4$ (Shanghai Aladdin Biochemical Technology Co., Ltd., Shanghai, China), 50.0 g/L $Kal(SO_4)_2$ (Shanghai Aladdin Biochemical Technology Co., Ltd., Shanghai, China), 10.0 g/L $(NH_4)_2MoO_4$ (Shanghai Aladdin Biochemical Technology Co., Ltd., Shanghai, China), 200 mL/L $H_2SO_4$ (Shanghai Aladdin Biochemical Technology Co., Ltd., Shanghai, China)) and 5 mL catalyst (8.8 g $Ag_2SO_4$/L $H_2SO_4$ (Shanghai Aladdin Biochemical Technology Co., Ltd., Shanghai, China)). Put the mixture in the XJ III digestion device (Shaoguan Tomorrow Environmental Protection Instrument Co., Ltd., Shaoguan, China), and set the temperature to 160 °C, with 25 min of digestion. The COD concentration is then determined in a spectrophotometer (visible spectrophotometer 752 N from Shanghai INESA Analytical Instrument Co., Ltd., Shanghai, China) set to 600 nm. The COD removal rate was calculated by Equation (2):

$$\mu = \frac{COD_{t0} - COD_{t7}}{COD_{t0}} \times 100\% \tag{2}$$

where μ is COD removal rate, $COD_{t0}$ is the initial COD concentration in the wastewater and $COD_{t7}$ is the 7th day COD concentration in the wastewater.

### 2.4.3. Impact of *Chlorella* sp. Concentrations on the COD Removal Rate

To quantify the relationship between the *Chlorella* sp. concentrations and COD removal rates, plot the curve of the *Chlorella* sp. concentrations and COD removal rates for the 7th day. The logarithm of the *Chlorella* sp. concentration was the x-axis, and the negative logarithm of the COD removal rate was the y-axis.

### 2.4.4. Impact of Extraction Concentrations on *Chlorella* sp. Proliferation Multiples

To quantify the relationship between the *Chlorella* sp. concentrations in BG-11 and proliferation multiples in the wastewater, plot the curve of the *Chlorella* sp. extraction concentrations and proliferation multiples for the 7th day. The *Chlorella* sp. extraction concentration from BG-11 was the x-axis, and the cell proliferation multiples in the wastewater were the y-axis. Origin 2023 was used for all data analysis.

## 3. Results and Discussion

### 3.1. Chlorella sp. Culture

In order to obtain *Chlorella* sp. cells and plot the growth curve of *Chlorella* sp., the *Chlorella* sp. was proliferated and cultured. In order to confirm the effect of air on the growth of *Chlorella* sp., the *Chlorella* sp. was cultured under two conditions: closed and open. After obtaining the growth curve of *Chlorella* sp., the precise concentration and proliferation factor were determined by diluting the *Chlorella* sp., and the standard concentration of the *Chlorella* sp. was accurate.

As shown in Figure 1A, the concentration of the *Chlorella* sp. in seal conditions was decreased; however, group I and group II in the open conditions were increased significantly from the 16th day. The concentrations of the *Chlorella* sp. increased faster in the open conditions by four to six times. It can be seen in Figure 1B,C that the proliferation multiples of *Chlorella* sp. varied between 1.00 and 2.00 (The proliferation multiples on the $i$th day were the ratio of the cell concentration on the $i$th day to day $i$-1th) during the process of the concentrations from $1.01 \times 10^6$ cell/mL and $2.82 \times 10^6$ cell/mL to $1.60 \times 10^7$ cell/mL. The difference between the two plots was due to the difference in the initial concentration of *Chlorella* sp. The higher the initial concentration, the faster the *Chlorella* sp. proliferates. There were maximums on the multiplier curve. The two peaks in group III corresponded to concentrations of $1.51 \times 10^6$ cell/mL and $4.50 \times 10^6$ cell/mL on the 3rd and 5th days (day $i$-1th), respectively. The peak in group IV corresponded to $6.97 \times 10^6$ cell/mL on the 3rd day.

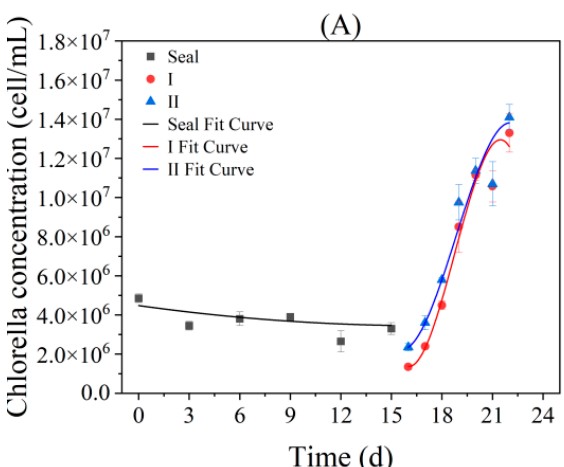
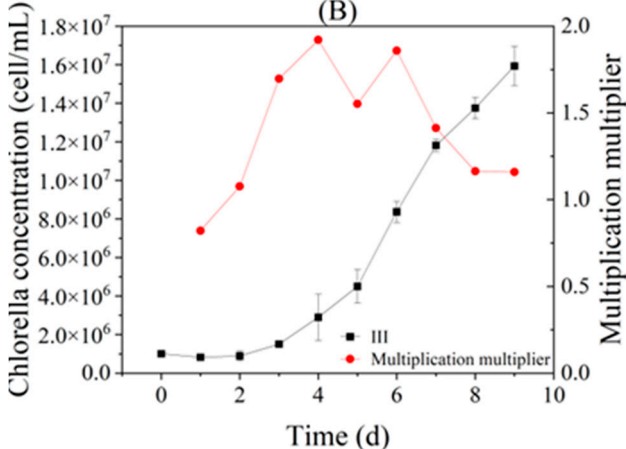

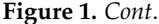

**Figure 1.** *Cont.*

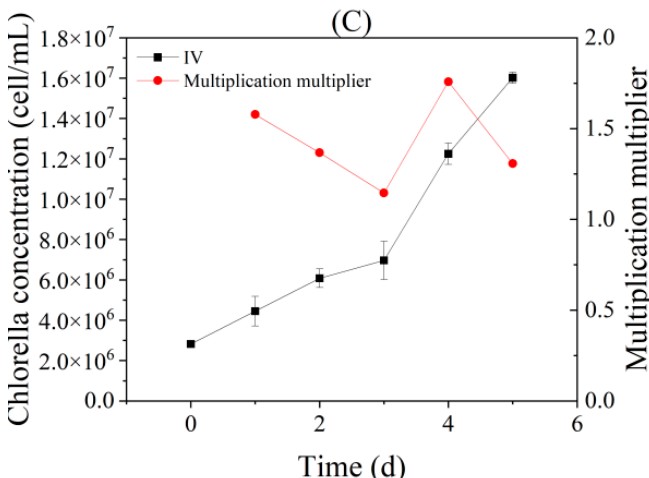

**Figure 1.** Culture of *Chlorella* sp. in BG-11 culture medium. (**A**) The changes in cell concentrations under the seal and open conditions. (**B**) The *Chlorella* sp. culture curve and the changes in the proliferation multiples at the $1.01 \times 10^6$ cell/mL initial concentration. (**C**) The *Chlorella* sp. culture curve and the changes in the proliferation multiples at the $2.82 \times 10^6$ cell/mL initial concentration.

Under the opening condition, the gas exchange between the air and the wastewater was sufficient, and this indicated that external oxygen and carbon dioxide [33] were required during the cultivation of *Chlorella* sp. The concentration of the *Chlorella* sp. raw solution was about $1.40 \times 10^7$ cell/mL. The concentration of $1.60 \times 10^7$ cell/mL with the proliferation multiples of 1.15–1.35 indicated that it was close to the stable stage. And the concentration of $1.60 \times 10^7$ cell/mL was used as the counting standard (100% ($v/v$)) for the *Chlorella* sp. According to the proliferation law of *Chlorella* sp., in order to proliferate rapidly, it is reasonable to ensure the initial concentration is between $4.00 \times 10^6$ cell/mL and $7.00 \times 10^6$ cell/mL for the continuous cultivation of *Chlorella* sp.

*3.2. Wastewater Treatment by Chlorella sp.*

In order to compare the effects of different *Chlorella* sp. dosages on COD removal, the dosages of the *Chlorella* sp. were controlled at 10%, 20% and 30% ($v/v$), respectively. The concentration change in the *Chlorella* sp. in COD wastewater and the concentration change in the COD in wastewater were determined, and the removal of COD was expressed by the concentration change in the COD.

The changes in the concentration of the *Chlorella* sp. with different initial dosages are shown in Figure 2A. The concentrations all increased rapidly in the first 3 days; then, all the groups gradually stabilized at different concentrations. Another study also showed that *Chlorella* sp. concentrations can stabilize in wastewater in 6 to 7 days [34]. The concentration of the 30% ($v/v$) group reached the maximum of $3.94 \times 10^7$ cell/mL on the 3rd day, and the concentration was $3.61 \times 10^7$ cell/mL on the 7th day. The maximum concentrations of the 20% ($v/v$) group and 10% ($v/v$) group were $2.62 \times 10^7$ cell/mL on the 5th day and $1.34 \times 10^7$ cell/mL on the 4th day, respectively, and the 7th day concentrations were $2.57 \times 10^7$ cell/mL and $1.15 \times 10^7$ cell/mL, respectively.

The changes in the COD concentrations and the COD removal rates with different initial *Chlorella* sp. dosages are shown in Figure 2B; the changes in the COD removal rates in the different groups had the same trend as the changes in the *Chlorella* sp. concentration. The cell concentrations and COD removal rates of the different groups increased with an increase in the initial dose of *Chlorella* sp. The initial COD concentration was 2000 mg/L. On the 7th day, the COD concentrations of the groups of 30%, 20% and 10% were 1087.5 mg/L, 1422.5 mg/L and 1787.5 mg/L, respectively, and the COD removal rates were 45.47%, 29.02% and 9.89%, respectively.

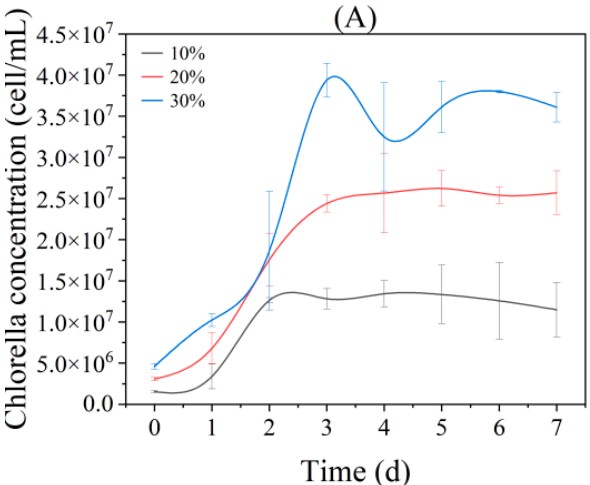
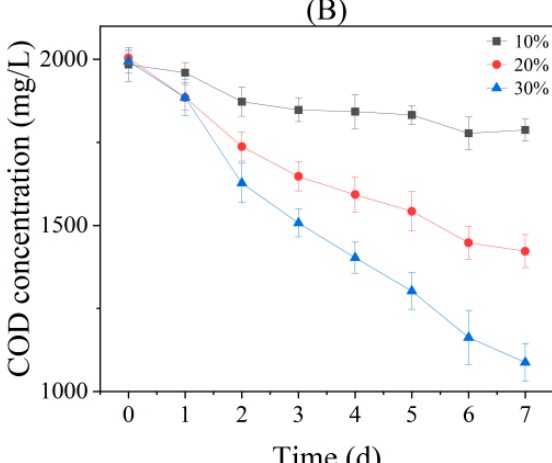

**Figure 2.** COD treatment by *Chlorella* sp. (**A**) The changes in the cell concentrations of *Chlorella* sp. in wastewater at dosages of 10%, 20% and 30% (*v/v*) *Chlorella* sp. (**B**) The changes in the COD concentrations in wastewater at dosages of 10%, 20% and 30% (*v/v*) *Chlorella* sp.

The concentrations obtained by the proliferation in COD wastewater were much greater than the standard concentration in BG-11. This is due to the large amount of organic carbon that promotes *Chlorella* sp. respiration and then provides material and energy for the proliferation process [32]. From 10% to 30%, the more *Chlorella* sp. added, the more new cells were proliferated, and more COD was removed. At the same time, the proliferation multiples increased from 7.48 to 8.37, then decreased to 7.84; with similar proliferation multiples, the added dosage determined the maximum concentration, and the COD was converted into the constituent substance of neonatal cells [19]. In the 20% and 30% groups, the cell concentrations on the 2nd day and the COD removal rates on the 1st day were similar, then immediately showed significant differences. The *Chlorella* sp. entered the adaptation phase in the COD solution within the first 2 days [35].

*3.3. Cell Activity and Proliferative Capacity of Chlorella* sp.

In order to analyze the effect of the concentration changes of the *Chlorella* sp. during the proliferation process in the BG-11 culture medium on the changes of the *Chlorella* sp. proliferation capacity, long-term concentration observation was performed on the cyclic culture process of the *Chlorella* sp. in two groups. And, by selecting different batches of *Chlorella* sp. for the control experiments, the changes in *Chlorella* sp. activity were analyzed.

The *Chlorella* sp. was grown and proliferated in the BG-11 culture medium, diluted after extraction and, then, the culture was continued and different batches of *Chlorella* sp. cells were obtained by cyclic operation. The changes in the concentrations of the *Chlorella* sp. in BG-11 medium are shown in Figure 3A,B. The maximum concentration of *Chlorella* sp. in BG-11 reached more than $4.00 \times 10^7$ cell/mL. The changes in the concentrations of the *Chlorella* sp. with different batches in wastewater are shown in Figure 3C. The cell concentrations of the four batches from B1 to B4 fluctuated between $2.07 \times 10^7$ cell/mL and $3.13 \times 10^7$ cell/mL on the 4th–7th days. The changes in the *Chlorella* sp. concentrations and the removal rates of COD with the different batches of *Chlorella* sp. on the 7th day are shown in Figure 3D. The COD removal rates of the four batches from B1 to B4 fluctuated between 24.00% (COD removal 480 mg/L) and 34.50% (COD removal 690 mg/L) on the 7th day, and the average was 28.88% (COD removal 577.6 mg/L).

After the *Chlorella* sp. adapted to the BG-11 environment, the greater the initial concentration, the more *Chlorella* sp. cells proliferated. However, BG-11 had limited nutrients, and the contents of various substances in the newborn cells were reduced. The concentrations of the four groups (B1, B2, B3 and B4) were significantly lower than the lowest concentration of $3.25 \times 10^7$ cell/mL of the initial 30% group on the 4th day.

The *Chlorella* sp. batches significantly influenced the final concentration of *Chlorella* sp. ($p < 0.05$). This indicated a decrease in the proliferative ability of *Chlorella* sp. in wastewater with the cyclic culture time. All the COD removal rates of the four groups decreased, and the removal rate of COD had a certain correlation with the concentration of the *Chlorella* sp. The *Chlorella* sp. batches significantly influenced the COD removal rate ($p < 0.05$). This indicated a decrease in *Chlorella* sp. cell activity and ability to remove COD. This may be due to some negative effects on the *Chlorella* sp. in the BG-11 medium. For the long-term culture, the supply of nutrients in BG-11 was insufficient, and the content of enzymes (N-containing) and genetic material (P-containing) in the cells decreased with the increase in *Chlorella* sp. concentration. This led to a decrease in cell activity and proliferative ability of *Chlorella* sp. At the same time, the secretions and excretions of *Chlorella* sp. cells in the culture medium increased, resulting in the deterioration of the solution environment, which is not conducive to the normal growth of cells. The activity of cell proliferation is directly related to various enzymes [36], and a long-term culture results in a change in the pH in the BG-11 solution, which will affect the life process of *Chlorella* sp. cells.

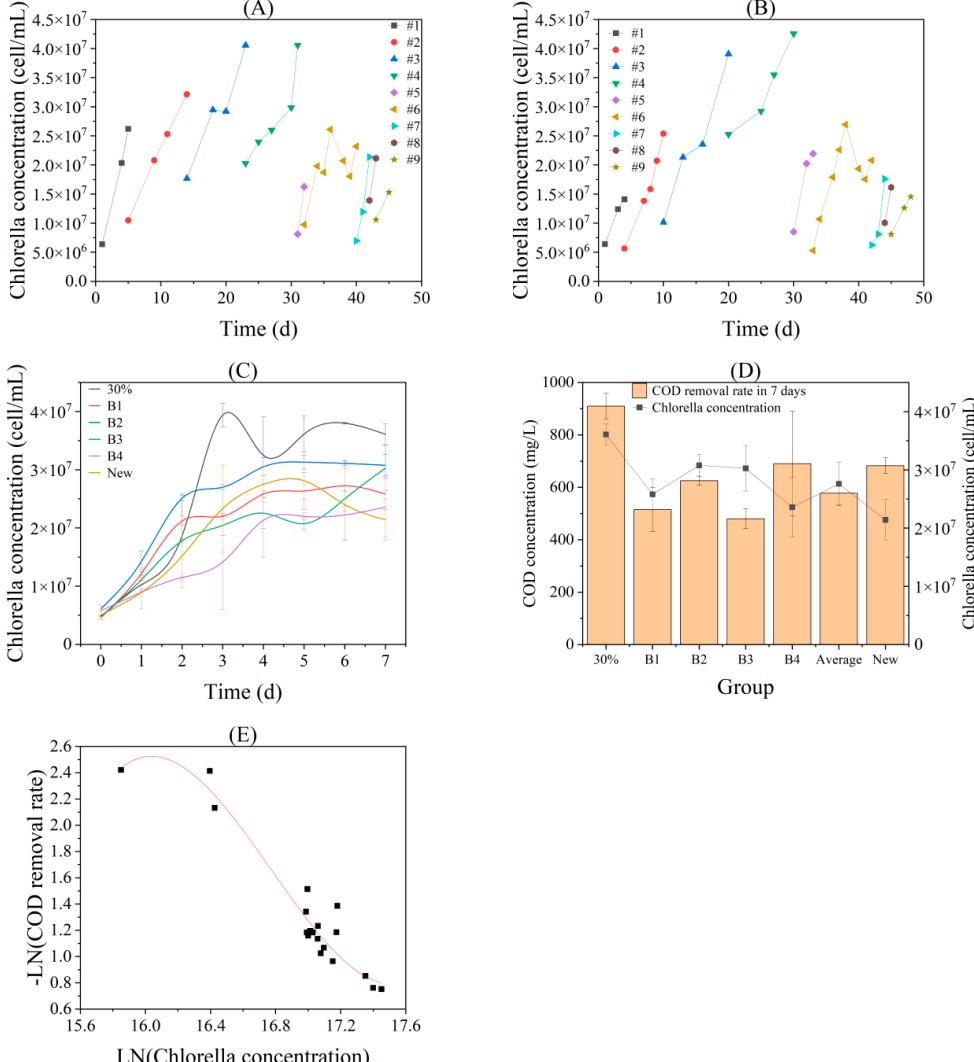

**Figure 3.** The changes in the *Chlorella* sp. concentration in BG-11 culture medium and wastewater, and the changes in the removal rate of COD. (**A**,**B**) The changes in concentrations of the *Chlorella* sp. in BG-11 during the cyclic culture process (from batch #1 to batch #9). (**C**) The changes in concentrations of the *Chlorella* sp. with different batches in wastewater. (**D**) The changes in the removal of COD in wastewater with different batches. (**E**) The relationship curve between the *Chlorella* sp. concentrations and the removal rates of COD on the 7th day.

The correlation of the *Chlorella* sp. concentrations and the removal rates of COD on the 7th day are shown in Figure 3E. The concentrations reflect the growth and proliferation ability of *Chlorella* sp., and the removal rates of COD represent the amount of organic matter absorbed by the *Chlorella* sp. As can be seen from Figure 3E, the COD removal rates of different groups increased with an increase in the final cell concentrations of the *Chlorella* sp. [32]. The formula for a regression curve plotted from scatter points is

$$y = -5052.39 + 907.34x - 54.19x^2 + 1.08x^3, R^2 = 0.9253.$$

The formula shows the correlation between *Chlorella* sp. concentrations and COD removal rates on the 7th day under the initial *Chlorella* sp. dosage from 10% to 30%. That is, the higher the *Chlorella* sp. concentration, the higher the removal rate of COD. The treatment of wastewater by *Chlorella* sp. can directly estimate the removal rate of COD, according to the formula.

The changes in the cell concentrations and the removal rate of COD on the 7th day in the wastewater treated by *Chlorella* sp. from the renewed BG-11 culture medium are shown in Figure 3C,D. The maximum cell concentration was $2.81 \times 10^7$ cell/mL, the final concentration was $2.14 \times 10^7$ cell/mL and the COD removal rate was 34.13% (COD removal 682.6 mg/L). After the cultivation in the renewed BG-11 culture medium, the proliferation capacity of *Chlorella* sp. in wastewater was recovered, but it was still not as good as the initial state. This indicated that, after multiple divisions of *Chlorella* sp. cells, the activity of the genetic material was irreversibly decreased, which was also consistent with the conjecture that the number of biological cell divisions is limited by telomeres [37,38].

The relationship of the *Chlorella* sp. concentrations in BG-11 culture medium and cells proliferation multipliers in wastewater on the 7th day is shown in Figure 4. The *Chlorella* sp. dosages in wastewater of all the groups were the same as the initial 30% group, but the concentrations of the *Chlorella* sp. extracted from BG-11 were different in different batches. The formula for a regression curve plotted from scatter points is

$$y = 13.372 - \left(4.626 \times 10^{-7}\right)x + \left(6.503 \times 10^{-15}\right)x^2, R^2 = 0.9000$$

**Figure 4.** The relationship of the *Chlorella* sp. concentrations in the BG-11 culture medium and cell proliferation multipliers in wastewater on the 7th day. The red curve is the fitted curve.

The cell proliferation multiplier in wastewater was the maximum when the *Chlorella* sp. concentration in BG-11 was $1.60 \times 10^7$ cell/mL, and then, the cells proliferation multipliers of different groups had decreased with an increase in concentrations of the *Chlorella* sp. in BG-11. But there is a limit to the descent. This suggested that $1.60 \times 10^7$ cell/mL as the standard concentration of *Chlorella* sp. can maintain cell activity. The six points of concentrations of the B4 group and renewed group were from $1.44 \times 10^7$ cell/mL to $2.26 \times 10^7$ cell/mL, but the proliferation multipliers were from 4 to 5. Neither changing the extraction concentrations nor renewing the culture medium could restore the proliferation ability of the *Chlorella* sp. This suggested that the proliferative activity of the *Chlorella* sp.

decreased irreversibly. This was determined by undesirable changes in the genetic material during cell division.

### 3.4. Effects of Natural Microorganisms

In order to visually observe and compare the changes in the microorganisms in COD wastewater, microscopic observation was used, and the state of the *Chlorella* sp. and natural microorganisms was represented by pictures.

In the process of wastewater treatment by *Chlorella* sp., natural microorganisms from the environment were introduced to the solution. The cells of the *Chlorella* sp. and natural microorganisms under the microscopes are shown in Figure 5. The natural microorganisms were much larger in volume than the *Chlorella* sp., could move, the solution color was earthy yellow and the cells were easy to aggregate and precipitate.

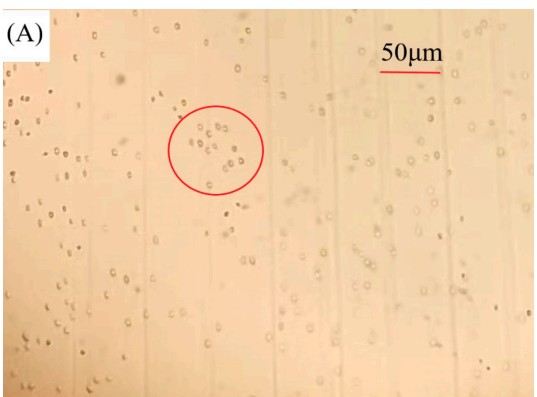
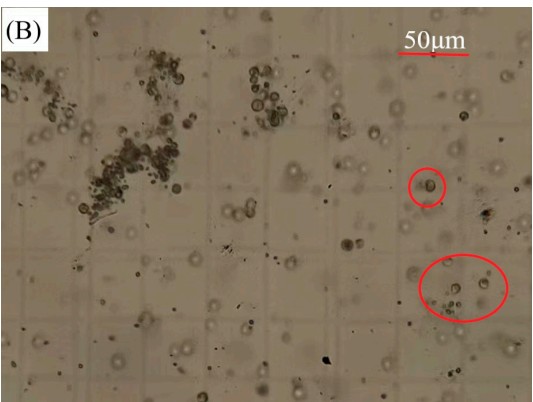

**Figure 5.** Cells under the microscopes. (**A**) *Chlorella* sp. (**B**) Natural microorganisms. Inside the red circle are the cells for focused observation. The red line is a 50 μm ruler length.

In order to compare the effects of natural microorganisms on the growth process of *Chlorella* sp. and the COD removal effect, the changes in the cell concentrations of the natural microorganisms and *Chlorella* sp. in the wastewater of the three groups were measured, and the COD removal was measured. The effect of the natural microorganisms on the *Chlorella* sp. concentration and the effect of COD removal were analyzed.

The three triangular bottles were recorded as A1, A2 and A3. The changes in the *Chlorella* sp. concentrations from the 0 to 7th day and the changes in the natural microorganism concentrations from the 8th to 25th day in three bottles of COD solution are shown in Figure 6A. From the 0 to 3rd day, the concentrations of the *Chlorella* sp. all showed a sharp upward trend. The cells in A2 began to enter a fluctuating state on the 3rd day, and the concentrations changed between $2.96 \times 10^7$ cell/mL and $3.23 \times 10^7$ cell/mL from the 3rd to 7th day, while A1 and A3 reached their maximum concentrations on the 4th day; the cell concentrations were $3.74 \times 10^7$ cell/mL and $4.34 \times 10^7$ cell/mL, respectively. But the *Chlorella* sp. concentrations of A1 and A3 decreased sharply and monotonously from the 3rd to 7th day, and the final concentrations were $1.00 \times 10^7$ cell/mL and $1.25 \times 10^7$ cell/mL, respectively. The rapid decrease in the *Chlorella* sp. concentrations in A1 and A3 was due to the presence of natural microorganisms on the 4th day; the natural microorganisms were observed under microscopes. The introduction of natural microorganisms led to contamination of the *Chlorella* sp. solution. Even though there were fewer natural microorganisms in A2, the *Chlorella* sp. were still affected during the long-term cultivation and the concentrations around $2.10 \times 10^7$ cell/mL on the 12th and 13th day. The concentrations of the natural microorganisms fluctuated between $3.00 \times 10^6$ cell/mL and $7.00 \times 10^6$ cell/mL from the 8th to 13th day. On the 21st day, $5.60 \times 10^6$ cell/mL was determined as the standard concentration; until the 25th day, the concentrations remained in a stable region.

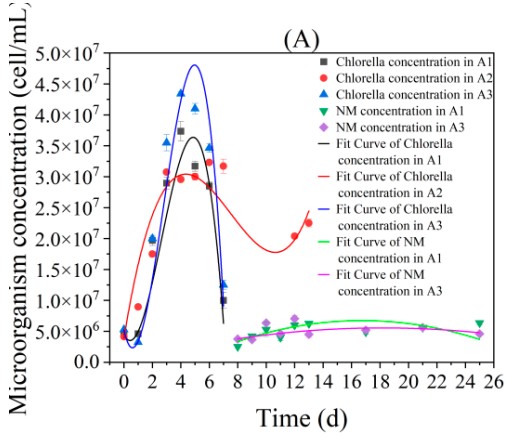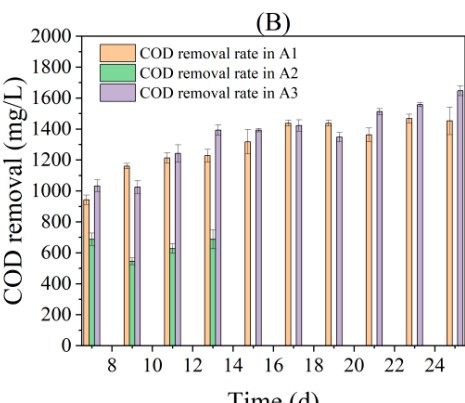

**Figure 6.** The influence of natural microorganisms on the *Chlorella* sp. and the treatment of wastewater. (**A**) The changes of *Chlorella* sp. concentrations in A1, A2 and A3 from the 0 to 7th day, and the changes of the natural microorganism (NM) concentrations in A1 and A3 from the 8th to 25th day. (**B**) The changes in the removal of COD of wastewater in A1, A2 and A3 from the 7th to 25th day.

The changes in the removal rates of COD in A1 and A3 from the 7th to 25th day and A2 from the 7th to 13th day are shown in Figure 6B. The COD removal rates of A1, A2 and A3 on the 7th day were 47.13%, 34.38% and 51.63% (COD removal 942.6 mg/L, 687.6 mg/L and 1032.6 mg/L), respectively. The removal rates of A1 and A3 gradually increased during the long-term treatment, and finally, removal rates of 72.62% (COD removal 1452.4 mg/L) and 82.38% (COD removal 1647.6 mg/L) were obtained on the 25th day, while the removal rates of A2 fluctuated between 27.25% (COD removal 545 mg/L) and 34.38% (COD removal 687.6 mg/L). This suggested that the presence of natural microorganisms was able to improve COD removal in the *Chlorella* sp. solution. Some studies have confirmed that other microorganisms appear when *Chlorella* sp. is cultured in wastewater, which may increase the removal rate [25,39,40].

### 3.5. Improvement of COD Removal with Natural Microorganisms

In order to determine the ability of natural microorganisms to remove COD alone, different concentrations of natural microorganisms were added to COD wastewater. By comparing the effects of COD removal, the ability of natural microorganisms to remove COD was analyzed, and the appropriate concentration was selected.

The concentration of $5.60 \times 10^6$ cell/mL was determined as the standard concentration for the natural microorganisms, and its mass concentration was between 1.25 g/L and 1.40 g/L. The changes in cell concentrations of the natural microorganisms at different dosages in different COD concentrations of wastewater are shown in Figure 7A,B. In the 2000 mg/L COD wastewater, the cell concentrations in three groups all increased from the 0 to 4th day but then all gradually decreased. The concentrations in the different groups increased with the increase in the initial dose of natural microorganisms. The maximums of 1%, 2% and 3% (*v/v*) were $1.80 \times 10^5$ cell/mL, $3.13 \times 10^5$ cell/mL and $4.83 \times 10^5$ cell/mL, respectively. All the data were less than 1/10 of the standard concentration. The cell concentrations in 1000mg/L COD were lower, and the maximums were $1.47 \times 10^5$ cell/mL, $2.23 \times 10^5$ cell/mL and $3.67 \times 10^5$ cell/mL, respectively. This indicated that natural microorganisms were difficult to proliferate alone in the wastewater containing COD only. The growth and proliferation of cells of natural microorganisms require elements such as N and P, and in wastewater containing only COD, natural microorganisms can only maintain simple life activities.

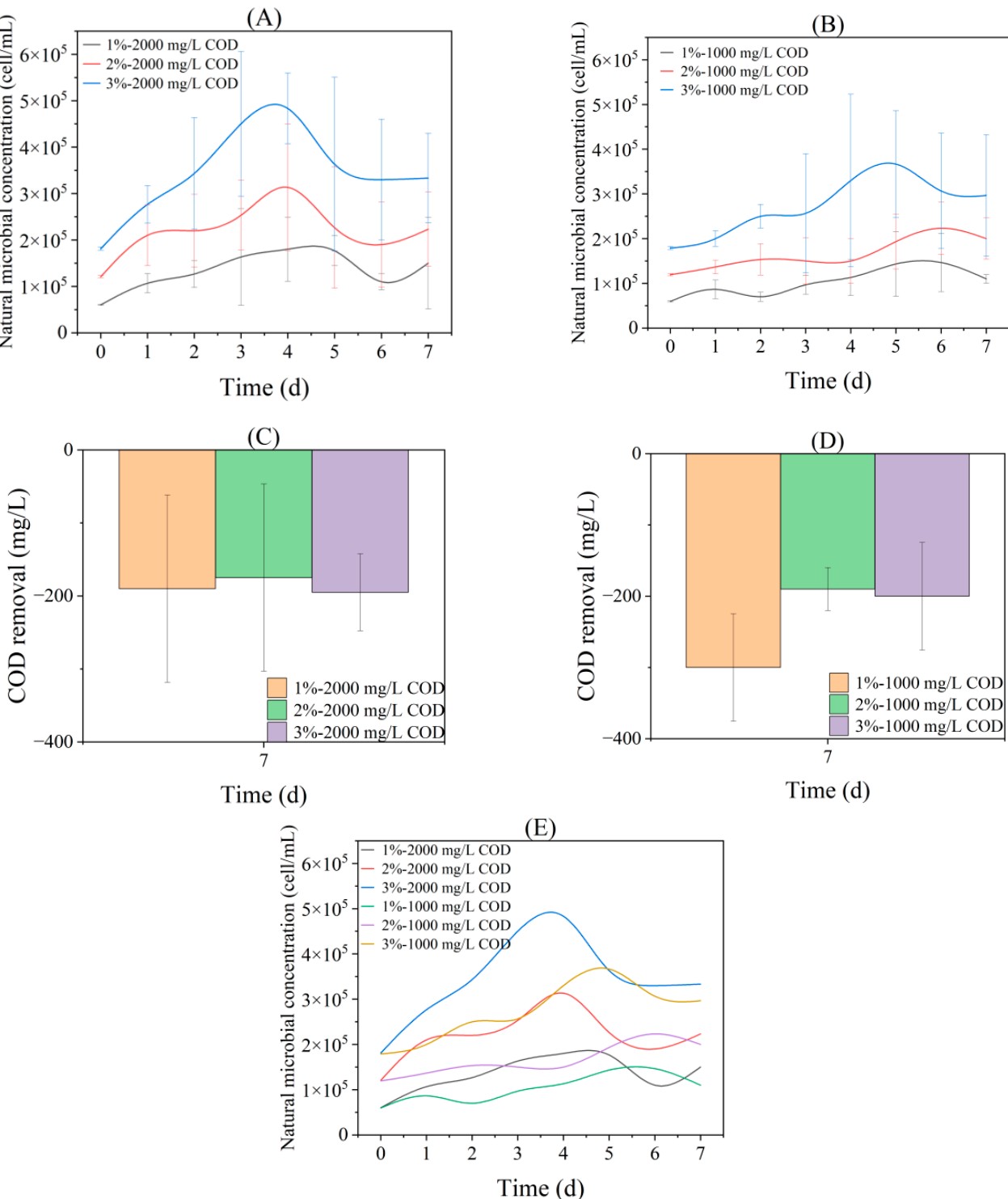

**Figure 7.** Treatment of COD by natural microorganisms alone. (**A**) The changes in cell concentrations with 1%, 2% and 3% (*v*/*v*) dosages in 2000 mg/L COD wastewater. (**B**) The changes in cell concentrations with 1%, 2% and 3% (*v*/*v*) dosages in 1000 mg/L COD wastewater. (**C**) The COD removal of 1%, 2% and 3% (*v*/*v*) dosages in 2000 mg/L COD wastewater on the 7th day. (**D**) The COD removal of 1%, 2% and 3% (*v*/*v*) dosages in 1000 mg/L COD wastewater on the 7th day. (**E**) The summary of the changes in the natural microorganism concentrations at different dosages in different concentrations of COD wastewater.

The COD removal rates of different dosages natural microorganisms in different COD concentrations of wastewater are shown in Figure 7C,D. The increase in the natural microorganism concentration was too small, and the consumption of COD was also too low. In the long-term treatment process, the phenomenon of wastewater evaporation concentration led to a negative COD removal rate.

A summary of the changes in the natural microorganism concentrations at different dosages in different COD concentrations of wastewater is shown in Figure 7E. All the data of both the maximum and the 7th day concentrations in descending order were 3%-2000 mg/L > 3%-1000 mg/L > 2%-2000 mg/L > 2%-1000 mg/L > 1%-2000 mg/L > 1%-1000 mg/L. This indicated that the natural microorganisms in wastewater with high concentrations of COD had a stronger proliferative ability. However, it was still far from the standard concentration.

In order to compare the effects of different *Chlorella* sp. dosages, natural microbial dosages and *Chlorella* sp. sources on the COD removal ability of the symbiotic system, the differences between different combinations were analyzed, and the mechanism was explored.

In the wastewater, the concentration of the natural microorganisms was 3% ($v/v$); the concentrations of the *Chlorella* sp. extracted from the BG-11 medium were 30% and 60% ($v/v$) and the concentrations of the *Chlorella* sp. extracted from the COD wastewater were 60%, 100% and 200% ($v/v$), respectively. The groups were recorded as 30% BG-3%, 60% BG-3%, 60%-3%, 100%-3% and 200%-3%, respectively.

The changes in *Chlorella* sp. concentrations in 2000 mg/L COD wastewater at different *Chlorella* sp. dosages and the 3% ($v/v$) natural microorganism dosage are shown in Figure 8A. Compared with the performance of *Chlorella* sp. alone, the addition of natural microorganisms significantly affected the normal proliferation process of *Chlorella* sp. The *Chlorella* sp. concentrations decreased in all groups and eventually converged in a small range. The *Chlorella* sp. in 30% BG-3% and 60% BG-3% were obtained from BG-11, and the concentrations decreased immediately on the 1st day but then fluctuated steadily. The *Chlorella* sp. in 60%-3%, 100%-3% and 200%-3% were obtained from the wastewater containing COD only, and the concentrations increased slightly on the 1st day but then dropped sharply. The *Chlorella* sp. concentrations of the five groups on the 7th day were $3.38 \times 10^6$ cell/mL, $6.55 \times 10^6$ cell/mL, $4.90 \times 10^6$ cell/mL, $4.77 \times 10^6$ cell/mL and $5.55 \times 10^6$ cell/mL, respectively. The initial concentration and source of the *Chlorella* sp. significantly influenced the final concentration of the *Chlorella* sp. ($p < 0.05$). All the *Chlorella* sp. concentrations of the different groups decreased in the presence of the natural microorganisms, but the *Chlorella* sp. cells did not disappear.

The changes in the natural microorganism concentrations in 2000 mg/L COD wastewater at different *Chlorella* sp. dosages and the 3% natural microorganism dosage are shown in Figure 8B. In all the groups, the concentrations of the natural microorganisms increased significantly. The concentrations of 60% BG-3% and 30% BG-3% were $6.77 \times 10^6$ cell/mL and $4.45 \times 10^6$ cell/mL, respectively; the data for the other groups were close. The initial concentration and source of the *Chlorella* sp. significantly influenced the final concentration of the natural microorganisms ($p < 0.05$). This suggested that the *Chlorella* sp. was necessary during the proliferation of natural microorganisms in the wastewater containing COD only, and the *Chlorella* sp. extracted from BG-11 had a stronger promoting effect.

The changes in the removal rates of COD in 2000 mg/L COD wastewater at different *Chlorella* sp. dosages and the 3% natural microorganism dosage are shown in Figure 8C. The COD removal rates of 60% BG-3% and 30% BG-3% increased rapidly in the first 3 days and started to fluctuate on the 4th day. On the 7th day, the COD removal rates of the 60% BG-3% group and 30% BG-3% group were 34.70% (COD removal 694 mg/L) and 45.75% (915 mg/L), respectively. The 45.75% of the 30%BG-3% group was the maximum, significantly higher than the average value of 28.88% of reduced activity and slightly higher than the 45.47% (909.4 mg/L) of the initial 30% group. The other groups increased slowly, especially 60%-3%, which fluctuated around 0%. The initial concentration and source of the *Chlorella* sp. significantly influenced the COD removal rate ($p < 0.05$). The *Chlorella* sp.

extracted from the BG-11 culture solution coexisted with the natural microorganisms to remove COD more effectively.

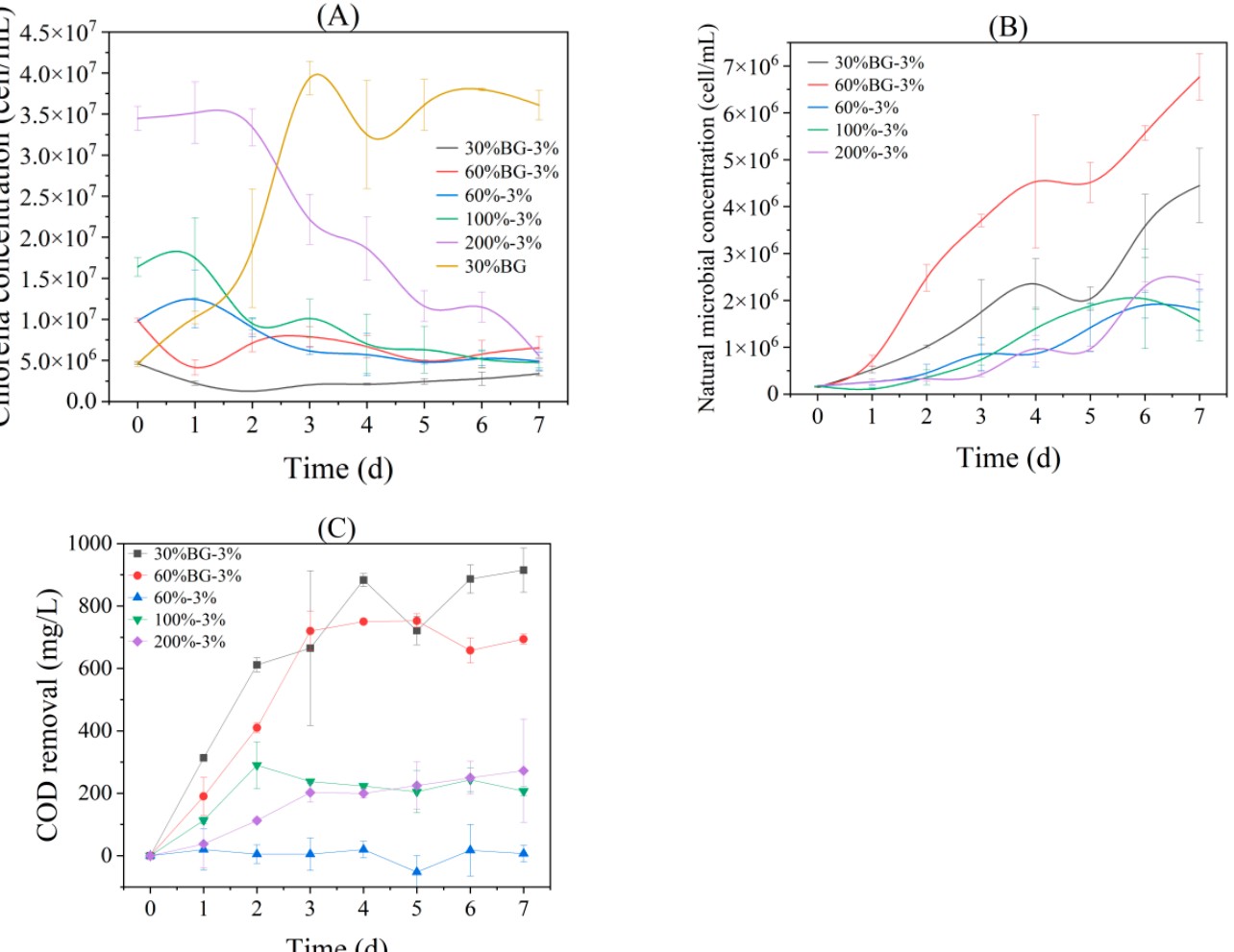

**Figure 8.** Treatment of wastewater by different dosages of *Chlorella* sp. and 3% (*v/v*) of natural microorganisms. (**A**) The changes in *Chlorella* sp. concentrations. (**B**) The changes in natural microorganism concentrations. (**C**) The changes in the removal of COD.

The changes in the concentrations of the *Chlorella* sp. and natural microorganisms led to changes in the removal rates of COD. *Chlorella* sp. absorbed $CO_2$ through photosynthesis [41] under light conditions, released $O_2$ and produced organic matter. The *Chlorella* sp. and natural microorganisms absorbed $O_2$ through respiration [42,43] and consumed COD to obtain energy for growth and proliferation and released $CO_2$. These processes occurred simultaneously in the *Chlorella* sp.–natural microorganism symbiotic system, where $O_2$ and $CO_2$ circulated between the two microorganisms. Increasing the $CO_2$ levels can promote *Chlorella* sp. photosynthesis, which may increase some of the activity of *Chlorella* sp. Endocytosis on *Chlorella* sp. by natural microorganisms was observed under microscopes. The relationship between the two microorganisms is predation [44,45]. This was in line with a trend that the concentrations of *Chlorella* sp. decreased while the concentrations of natural microorganisms increased, but there were limit values, and they eventually stabilized. The *Chlorella* sp. cells were rich in various proteins [25], lipids [16] and elements, which were necessary for the growth and proliferation of the natural microorganisms.

*Chlorella* sp. extracted from BG-11 maintained a high content of cellular inclusions, including genetic material (containing P) and proteins (containing N). The concentrations of the *Chlorella* sp. were controlled in the logarithmic growth stage under the influence of

natural microorganisms. In this condition, the *Chlorella* sp. maintained a high proliferative activity and provided new cells [32] to natural microorganisms for a long time. The natural microorganisms preyed on highly nutrient-rich *Chlorella* sp. and proliferated in large numbers. In this case, the larger the dosage of *Chlorella* sp., the more natural microorganisms were obtained on the 7th day. The respiration of *Chlorella* sp. and natural microorganisms was the main part of COD consumption. However, too much $CO_2$ from too many natural microorganisms had a negative impact on the symbiotic system stability and wastewater treatment efficiency, which may be the reason why the COD removal rate of 30% BG-3% was better than 60% BG-3%. In a word, the optimal dosage of *Chlorella* sp. was 30% (*v*/*v*) from BG-11.

Elements such as N, P and others have a direct impact on the development of microbial communities [46]. The *Chlorella* sp. extracted from the wastewater containing COD only, which underwent 7 days of growth and proliferation without a supply of N and P, resulted in extremely low levels of genetic material and protein in the cells. The proliferative ability was seriously decreased, and it was impossible to maintain the cell concentrations under the influence of the natural microorganisms. Studies have shown that the degradation of some specific organic matter by microorganisms is achieved through specific enzymes, and the content of N directly affects the removal effect of pollutants [47,48]. At the same time, it is difficult for *Chlorella* sp. with large concentrations but few nutrients to supply the proliferation needs of natural microorganisms. This leads to the slow growth of natural microorganism concentrations. In this case, the COD was consumed by natural microorganisms absorbing organic matter for the energy for endocytosis [49] *Chlorella* sp., due to the supply of organic matter in *Chlorella* sp. cells and the small concentrations of natural microorganisms, there was less demand for natural microorganisms for organic matter in wastewater, which led to smaller COD removal rates.

In the COD wastewater, the concentrations of natural microorganisms were 1%, 3% and 6% (*v*/*v*), respectively. The *Chlorella* sp. in 30% BG-6%(1) was extracted from an excessive concentration in BG-11; the others were all extracted from the standard concentration in BG-11.

The changes in *Chlorella* sp. concentrations in 2000 mg/L COD wastewater at the 30% (*v*/*v*) *Chlorella* sp. dosage and different natural microorganism dosages are shown in Figure 9A. The *Chlorella* sp. concentrations in all the groups fluctuated, and the final concentrations were less than the initial data. The initial concentration of natural microorganisms did not significantly influence the final concentration of *Chlorella* sp. ($p < 0.05$). This indicated that *Chlorella* sp. can maintain the concentration of *Chlorella* sp. under the influence of natural microorganisms without being completely removed.

The changes in the natural microorganism concentrations in 2000 mg/L COD wastewater at the 30% (*v*/*v*) *Chlorella* sp. dosage and different natural microorganism dosages are shown in Figure 9B. The 30% BG-6%(2) group obtained the maximum natural microorganisms concentration of $5.00 \times 10^6$ cell/mL, but the 30% BG-6%(1) group had a similar concentration as the 30% BG-1% group. The initial concentration of the natural microorganisms significantly influenced the final concentration of the natural microorganisms ($p < 0.05$). Under the same conditions, the concentration of the natural microorganisms increased with an increase in the initial dosage, but there were also limit values.

The changes in the removal rates of COD in 2000 mg/L COD wastewater at the 30% (*v*/*v*) *Chlorella* sp. dosage and different natural microorganism dosages are shown in Figure 9C. The initial concentration of the natural microorganisms significantly influenced the COD removal rate ($p < 0.05$). Neither too large nor too few dosages of natural microorganisms could achieve high COD removal rates. The optimal dosage of natural microorganisms was 3% (*v*/*v*). In the symbiotic system, the optimal combination was 30% (*v*/*v*) *Chlorella* sp. from BG-11 and 3% (*v*/*v*) natural microorganisms.

In the process of predation, the relatively more biomass of the predator, the faster its concentration increased, but there were limit values. The 30% BG-6%(2) group obtained the maximum concentrations of *Chlorella* sp. and natural microorganisms but with the COD removal rate less than 30% (COD removal 600 mg/L). The large amount of $CO_2$ pro-

duced by natural microorganisms provided enough raw materials for the photosynthesis of *Chlorella* sp. and increased the content of organic matter in the *Chlorella* sp. cells. This reduced the demand of natural microorganisms for the *Chlorella* sp. cells and COD; therefore, the concentrations of the *Chlorella* sp. cells and COD in wastewater were maintained at a high level. The 30% BG-6%(1) group had a larger dosage of natural microorganisms, but the final concentration was similar to the 30% BG-1% group, and the COD removal rate showed the same performance. The *Chlorella* sp. in 30% BG-6%(1) was extracted from an excessive concentration in BG-11. Multiple cell divisions resulted in relatively low cell activity and proliferative ability of the *Chlorella* sp., and the cells had low levels of nutrients. Therefore, even with a large initial dosage of natural microorganisms, without a supply of high-quality *Chlorella* sp., the wastewater treatment could not get the expected effect.

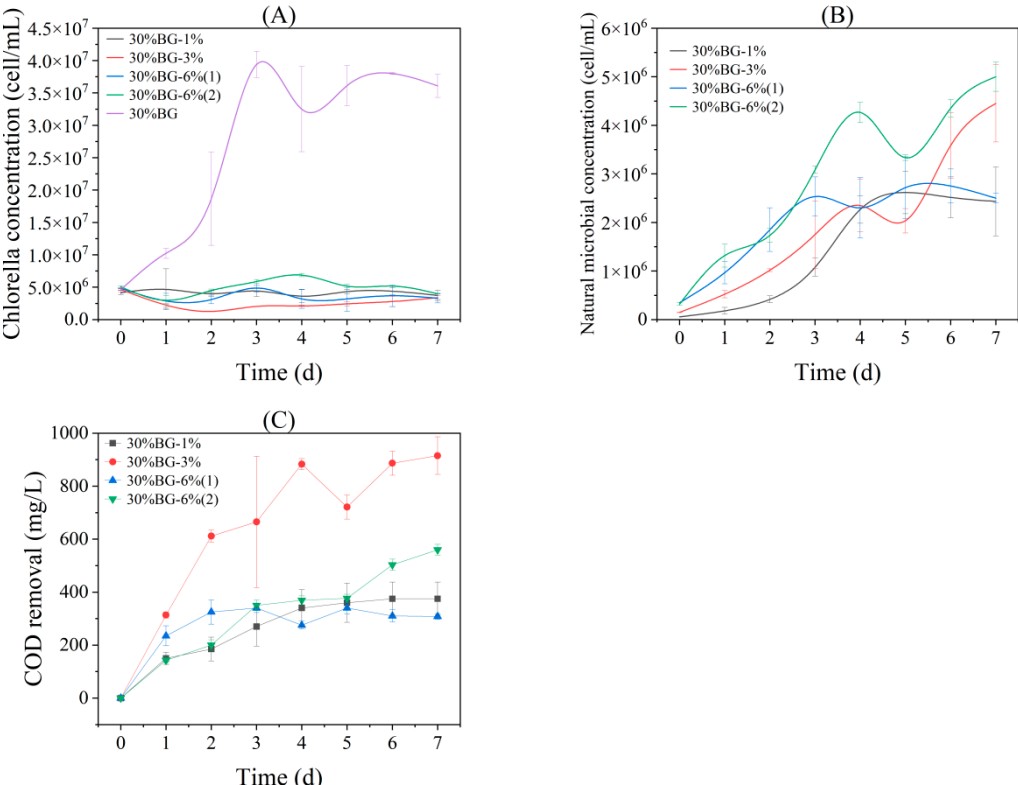

**Figure 9.** Treatment of wastewater by 30% (*v/v*) of *Chlorella* sp. and different dosages of natural microorganisms. (**A**) The changes in *Chlorella* sp. concentrations. (**B**) The changes in natural microorganism concentrations. (**C**) The changes in the removal of COD.

The changes in *Chlorella* sp. concentrations in different dosage combinations of *Chlorella* sp. and natural microorganisms in wastewater are shown in Figure 10A. In all different combinations, the concentrations of *Chlorella* sp. decreased significantly. Finally, the maximum concentration was only $6.55 \times 10^6$ cell/mL, which is 40.94% of the standard concentration in BG-11 and 16.62% of the maximum concentration in wastewater. This suggested that natural microorganisms can effectively control *Chlorella* sp. concentrations. Natural microorganisms can be applied to remove *Chlorella* sp. cells in wastewater.

The changes in natural microorganism concentrations in different dosage combinations of *Chlorella* sp. and natural microorganisms in wastewater are shown in Figure 10B. In all combinations, the standard concentration was hardly exceeded, and there was no pollution of excess natural microorganism cells.

The changes in the removal rates of COD in different dosage combinations of *Chlorella* sp. and natural microorganisms in 2000mg/L COD wastewater are shown in Figure 10C. Natural microorganisms preyed on *Chlorella* sp. while consuming COD in wastewater. A

change in the ratio of *Chlorella* sp. to natural microorganisms can improve the COD removal rate of the system.

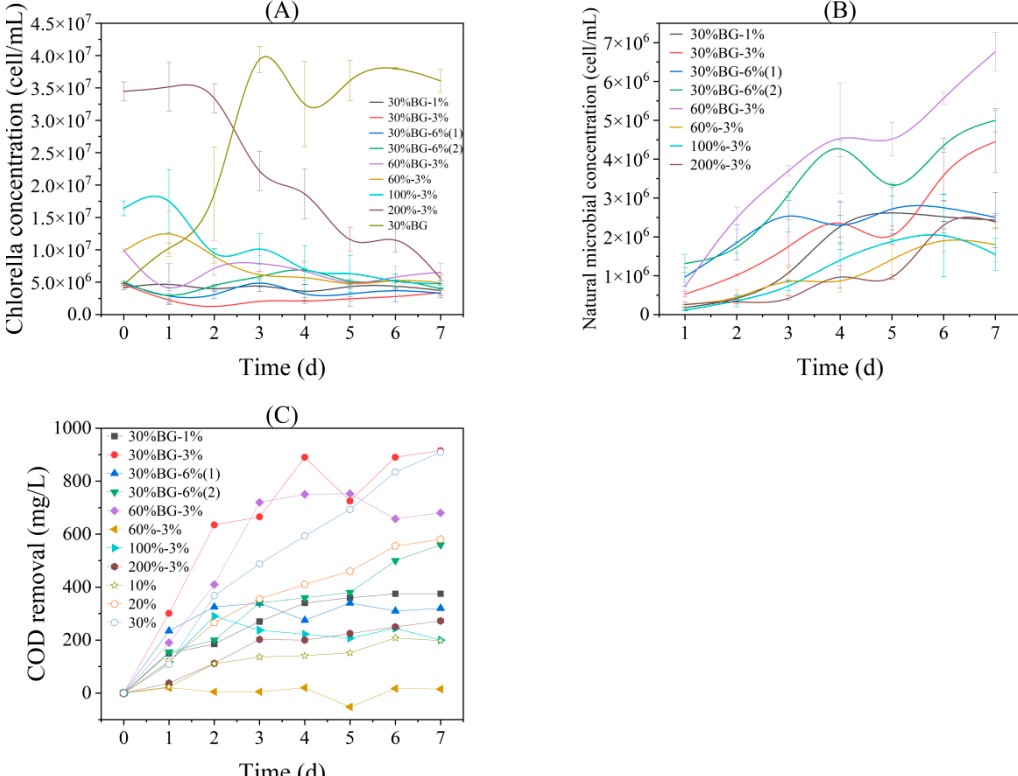

**Figure 10.** Summary of different dosage combinations of *Chlorella* sp. and natural microorganisms in wastewater treatment. (**A**) The changes in *Chlorella* sp. concentrations. (**B**) The changes in natural microorganism concentrations. (**C**) The changes in the removal of COD.

The optimal combination, the 30% BG-3% group, obtained a COD removal rate slightly higher than the initial 30% *Chlorella* sp. group by promoting *Chlorella* sp. cell activity and maintaining the *Chlorella* sp. proliferative ability, and the final *Chlorella* sp. concentration was only 9.38% of the initial 30% *Chlorella* sp. group. The addition of natural microorganisms solved the problem of reduced activity of the *Chlorella* sp. and the excessive concentration of *Chlorella* sp. This may be applied to the treatment of red tides [30].

In order to compare the effects of different factors on the symbiotic system, the *Chlorella* sp. concentration, natural microbial concentration, *Chlorella* sp. removal and COD removal on the 7th day were compared, and the main determinants of COD removal in the symbiotic system were analyzed.

On the 7th day, the concentrations of *Chlorella* sp., the concentrations of natural microorganisms, the COD removal rates and the removal rates of *Chlorella* sp. in different groups are shown in Figure 11A. In different groups, *Chlorella* sp. extracted from BG-11 can obtain higher COD removal rates with fewer dosages. This was mainly due to the growth and proliferative activity of *Chlorella* sp. cells. With the 30% *Chlorella* sp. dosage, the larger the natural microbial dosage, the greater the removal rate of *Chlorella* sp., but the final concentrations of *Chlorella* sp. were close to each other. With the 60% *Chlorella* sp. dosage, the cells extracted from BG-11 can maintain the concentration; therefore, the 60%-3% group obtained a higher removal rate of *Chlorella* sp. The three groups with *Chlorella* sp. extracted from wastewater containing COD only were 60%-3%, 100%-3% and 200%-3%. In these groups, the *Chlorella* sp. cells almost lost the ability to grow and proliferate, as predation dominated in the system. Since the final *Chlorella* sp. concentrations were close, the larger the *Chlorella* sp. dosage, the greater the removal rate of *Chlorella* sp., and the *Chlorella* sp. removal rates of 60%-3%, 100%-3% and 200%-3% were 50.06%, 70.79% and

83.86%, respectively. All the *Chlorella* sp. concentrations in the different groups were below $6.55 \times 10^6$ cell/mL, which was 40.94% of the standard concentration ($1.60 \times 10^7$ cell/mL) and 18.14% of the 30% group on the 7th day concentration ($3.61 \times 10^7$ cell/mL). This indicated that natural microorganisms had a superior effect on the cell concentration control of *Chlorella* sp. under different conditions. Natural microorganisms can also be used to treat algae pollution caused by *Chlorella* sp. [50].

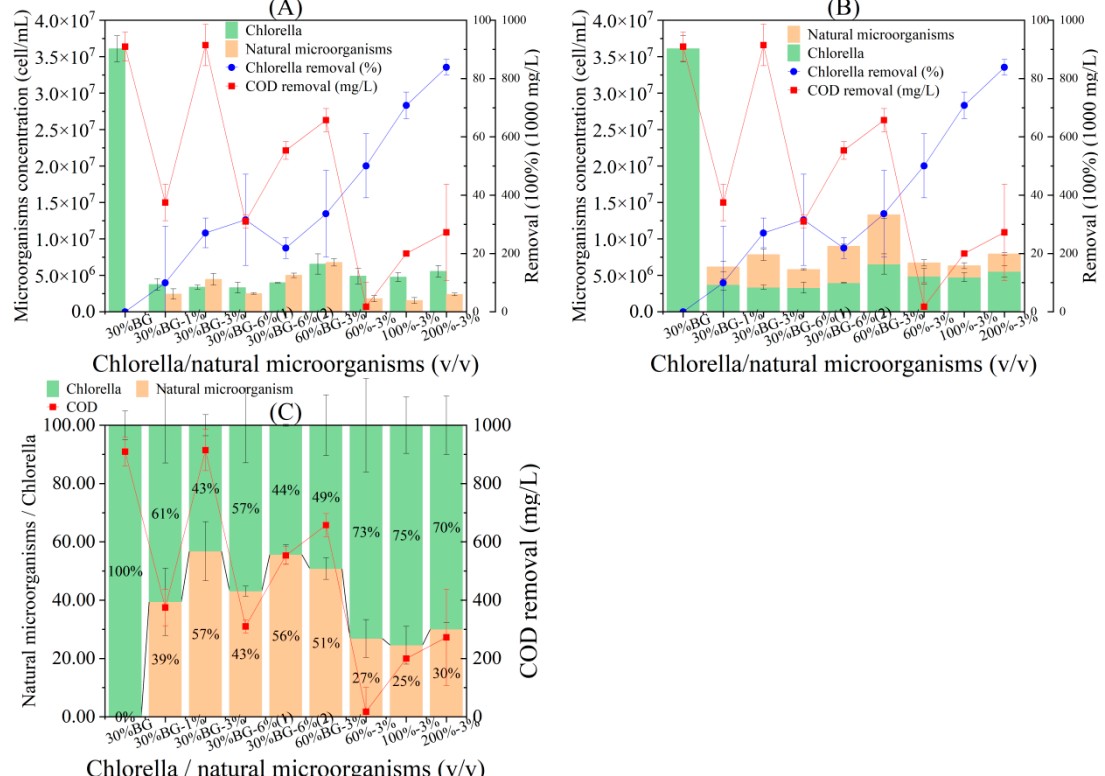

**Figure 11.** The effects of the *Chlorella* sp. concentration and natural microorganism concentration on the removal rate of COD and the removal rate of *Chlorella* sp. (**A**) The effects of the *Chlorella* sp. concentrations and natural microorganism concentrations. (**B**) The effects of the total microorganism concentrations. (**C**) The effects of the proportion of *Chlorella* sp. and natural microorganisms in the total microorganism concentrations.

On the 7th day, the total concentrations of the *Chlorella* sp. and natural microorganisms, the removal rates of COD and the removal rates of the *Chlorella* sp. in the different groups are shown in Figure 11B. The high total microorganism concentrations promoted the COD removal rates. This was due to the fact that the respiration of microorganisms is the main way to consume COD; the more cells, the stronger the respiration of the system. The 60%-3% group was a special case, with a COD removal rate close to 0%. This was due to the small dosage and low cell activity of the *Chlorella* sp., leading to the contribution of the *Chlorella* sp. to COD consumption being extremely low. At the same time, the *Chlorella* sp. cells had almost no new supplements; that is, natural microorganisms can only prey on the original low-quality *Chlorella* sp., and less COD was consumed during predation.

On the 7th day, the proportion of the *Chlorella* sp. and natural microorganisms in the total microorganism concentrations and the removal rates of COD are shown in Figure 11C. The removal rates of COD had a certain correlation with the proportion of natural microorganism concentrations in the total microorganism concentrations. Natural microorganisms clearly dominated the treatment of COD in the symbiotic systems.

## 4. Conclusions

In the study, a *Chlorella* sp.–natural microorganism symbiotic system was prepared for COD treatment. A long-term culture led to a decrease in the nutrient content in *Chlorella* sp. cells, resulting in a decrease in the cell activity, proliferation capacity and COD removal ability of *Chlorella* sp. The symbiotic system promoted *Chlorella* sp. activity by facilitating the circulation of $O_2$ and $CO_2$ between *Chlorella* sp. and natural microorganisms. Natural microorganisms controlled the concentration of *Chlorella* sp. near the logarithmic growth stage, which can maintain the optimal proliferation ability of *Chlorella* sp. for a long time. The removal rate of COD increased from 28.88% to 45.75%, and the concentrations of the *Chlorella* sp. in all the groups were controlled below $6.55 \times 10^6$ cell/mL by predation. The symbiotic system can simultaneously improve the COD removal rate and limit *Chlorella* sp. concentrations.

**Author Contributions:** Conceptualization, Q.S. and X.Z. (Xiaoping Zhang); methodology, Q.S. and X.Z. (Xiaoping Zhang); software, Q.S.; validation, Q.S.; formal analysis, Q.S. and X.Z. (Xiaoping Zhang); investigation, Q.S.; resources, Q.S.; data curation, Q.S.; writing—original draft preparation, Q.S.; writing—review and editing, Q.S., X.Z. (Xiaoping Zhang) and X.Z. (Xin Zhang); visualization, Q.S.; supervision, Q.S. and X.Z. (Xiaoping Zhang); project administration, X.Z. (Xiaoping Zhang) and funding acquisition, X.Z. (Xiaoping Zhang). All authors have read and agreed to the published version of the manuscript.

**Funding:** This research was funded by the National Key Research and Development Program of China (Grant No. 2016YFC0400702-2). The Guangdong Science and Technology Program (2020B121201003).

**Data Availability Statement:** The data presented in this study are available on request from the corresponding author. The data are not publicly available due to [privacy].

**Conflicts of Interest:** The authors declare no conflict of interest. The funders had no role in the design of the study; in the collection, analyses or interpretation of the data; in the writing of the manuscript or in the decision to publish the results.

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
