# Peer review of "Impact of Natural Microorganisms on the Removal of COD and the Cells Activity of the Chlorella sp. in Wastewater"

_water, doi:10.3390/w15203544_

Round 1

Reviewer 1 Report

 This important study proposes the COD removal kinetics observation of wastewater containing only COD by Chlorella sp, but introduction is unclear what organisms were used. The cells activity and proliferation ability of Chlorella sp. decreased with the culture time, which affected the removal of COD in wastewater. Chlorella sp.-natural microorganism symbiosis system was prepared- elaborate more consisting system and revise wording.

There was explored how the natural microorganisms affect the cell activity and the proliferation ability of Chlorella sp. in wastewater. In the treatment of COD by Chlorella sp., the removal rate of COD decreased from 45.47% to 28.88% with a decrease in cell activity and proliferation ability of Chlorella sp.. In the Chlorella sp.-natural microorganism symbiotic system, the removal rate of COD reached 45.75%.- Give standard deviations

 With the introduction of natural microorganisms, the circulation of CO2 and O2 between Chlorella sp. and natural microorganisms promoted photosynthesis and respiration, which enhanced the cell activity of Chlorella sp.. Under the condition that the dosage of natural microorganisms was between 1% and 6%, the concentration of Chlorella sp. was close to the logarithmic growth phase, which maintained the proliferation ability of Chlorella sp.. At the same time, the natural microorganisms grew and proliferated in wastewater containing only COD through preying on Chlorella sp.

Further notes could be made on MS quality:

1. Figures axis titles are very small as well fonts need to be enhanced and unified proper style. Legend of Figs is unclear and not understandable and self explanatory, revise.

2. Results section needs broader introduction, thereafter can start specific results.

3. "According to the proliferation law of Chlorella sp., in order to proliferate rapidly, it is reasonable to ensure the initial concentrations is between 4.00×10 6 cell/mL and 7.00×10 6 cell/mL for continuous cultivation of Chlorella sp.- elaborate why this concentration is optimum, some reference to prove it.

4. COD removal rate is not in %, but in mg L day.

5.  Control tests are missing as well.

6.  Fig. 2 legend should be inside Graph.  COD removal rate is not in %, but in mg L day.

7. Why no significant difference p values are given based on changes in conditions did not occur as per paper and are not depicted by exact p value?

8. Data, text and Figures could include more standard deviation and be higher quality.

9. Figures sharpness needs to be adjusted. Numeric results needed to be added to results.

10. For preparation and new water treatment, it has been successfully done earlier and involved systems have been effectively characterized:    https://doi.org/10.3390/w13212959 https://doi.org/10.3390/w1413206 https://doi.org/10.3176/oil.2017.1.06 https://doi.org/10.1007/s13762-023-05055-9

11. The main parameters affecting the efficiency of the dye treatment, such as pH, the mass of biomass added to the solution, the concentration of DO supplied should be included in discussions.

12. All figures are foggy.

There is needed qPCR or pyrosequencing results confirming microbiological composition of biomass and its actual quantities and prove pollution removal and chlorella removal.

13. R598  "The symbiotic system promoted Chlorella sp. activity by  facilitating the circulation of O2 and CO2 between Chlorella sp. and natural microorganisms. - revise to add O2 and CO2 measures to prove Your statement.

14. The Erlenmeyer flasks were shaken evenly twice a day- why not continuously, how DO and pH change and stable values were maintained in this technique?

15. Parallel tests results and nr of parallels and controls needed to be added, language quality of MS checked and statistics done.

16. "However, BG-11 had limited nutrients, and the content of various substances in the newborn cells may be reduced".- text not understood

Need revision

Reviewer 2 Report

This study investigated effect of microbes on COD removal and Chlorella sp.  

1. How did the concentration of Chlorella sp. change in the wastewater when exposed to different Chlorella sp. dosages and 3% natural microorganism dosage? Were there significant differences between Chlorella sp. extracted from BG-11 Medium and COD wastewater in terms of their growth and concentration changes?

2. Can you elaborate on the relationship between the concentrations of Chlorella sp. and natural microorganisms and how it impacted the removal rates of COD in the wastewater? What role did Chlorella sp. play in promoting the proliferation of natural microorganisms, and did the source of Chlorella sp. influence this?

3. What factors influenced the choice of the optimal Chlorella sp. dosage (30% v/v from BG-11) for efficient wastewater treatment? How did the nutrient content in Chlorella sp. extracted from COD wastewater affect its proliferative ability, and why did it result in lower COD removal rates compared to Chlorella sp. from BG-11?

4. How does long-term culture affect the nutrient content and activity of Chlorella sp. cells in the Chlorella sp.-natural microorganism symbiotic system, and what are the consequences for the system's COD removal capacity?

5. Can you explain the role of natural microorganisms in controlling the concentration of Chlorella sp. in the symbiotic system and how it helps maintain Chlorella sp.'s optimal proliferation ability? What is the significance of this control for COD removal?

6. How does the symbiotic system enhance COD removal rates, and how does it simultaneously limit the concentration of Chlorella sp.? What are the practical implications of this dual effect for wastewater treatment applications?

Minor comments,

In all the graphs, it is difficult to discern the text on both the x-axis and y-axis. It seems that either enlarging the text or increasing the size of the figures may be necessary.

Round 2

Reviewer 2 Report

There was no special comments